# Embedding Performance into Decision Logic: A KPI-Driven Framework for Omnichannel Logistics Networks under Uncertainty

**Zhang Wenye**
Graduate School of Service and Trade
Peter the Great St. Petersburg Polytechnic University
St. Petersburg, Russia
ZhangWenye@yandex.ru

**Sergey Evgenievich Barykin**
Graduate School of Service and Trade
Peter the Great St. Petersburg Polytechnic University
St. Petersburg, Russia
barykin_se@spbstu.ru

## Abstract

Decision-making in omnichannel logistics networks requires balancing multiple performance dimensions under dynamic and uncertain operating conditions. In practice, key performance indicators (KPIs) are frequently applied for ex-post evaluation, while their direct integration into decision formulation remains limited, leading to a structural separation between performance management and alternative selection. This study proposes a KPI-driven decision framework that embeds multidimensional performance indicators at the modeling stage. Alternatives are represented within a unified multi-criteria structure, where KPI dimensions are normalized and aggregated into a weighted composite index. The decision problem is formulated as the maximization of expected composite performance over discrete alternatives under scenario uncertainty. A simulation-based operationalization is developed to incorporate stochastic variation in key indicators while preserving a consistent comparison logic across scenarios. The resulting structure enables robust ranking and selection of alternatives based on expected multidimensional performance. The study provides an interpretable and computationally reproducible framework for multi-indicator decision support in omnichannel logistics networks.

## 1 Introduction

Omnichannel logistics networks (OLNs) represent a structural transformation of contemporary distribution systems. Rather than functioning as linear transportation chains, they operate as coordinated multi-layer configurations in which physical flows, information streams, and financial transactions are integrated within a unified service architecture. Recent studies have proposed digital infrastructures that integrate material, financial, and informational flows through hypernetwork architectures and blockchain-synchronized digital twin systems, enabling real-time coordination in complex logistics and urban ecosystems (Ruponen et al., 2025). This structural coupling creates decision environments characterized by multidimensional performance interdependencies and dynamic uncertainty. Decisions in such networks require balancing economic efficiency, service quality, environmental sustainability, and operational stability across interconnected layers.

From a theoretical standpoint, OLNs can be interpreted as performance-embedded decision systems. Unlike traditional logistics models that treat performance indicators as external evaluation criteria, omnichannel environments require that performance dimensions be structurally integrated into the

configuration logic of alternatives. In this sense, decision-making is not merely a problem of parameter optimization but a problem of structuring a multidimensional decision space under uncertainty.

Existing research on logistics decision-making has developed along two dominant trajectories. The first trajectory emphasizes solver-based optimization, typically employing mathematical programming and operations research techniques to design networks, allocate facilities, or determine routing strategies. These approaches provide analytical rigor and often focus on minimizing cost or maximizing profit subject to operational constraints. However, performance indicators in such models are usually embedded as coefficients or constraints rather than conceptualized as structural dimensions of the decision space itself.

The second trajectory focuses on performance management and the development of key performance indicators (KPIs) to measure logistics outcomes. Performance measurement frameworks facilitate monitoring, benchmarking, and managerial control. With the advancement of multi-criteria decision-making (MCDM) techniques, researchers have attempted to compare alternatives across multiple performance dimensions and to incorporate uncertainty through scenario analysis or simulation. Yet, even within these approaches, KPIs are frequently employed as ex-post evaluation tools. They assess predefined alternatives rather than define the architecture of the decision process.

Accordingly, a methodological discontinuity persists between performance logic and decision logic. Performance measurement systems describe outcomes, whereas decision models structure choices; the two are rarely integrated at the foundational modeling stage. This separation becomes increasingly problematic in omnichannel logistics networks, where interdependencies across layers amplify the effects of multidimensional trade-offs and uncertainty.

To address this theoretical gap, the present study advances a KPI-embedded decision framework that reconceptualizes performance indicators as constitutive elements of the decision space. In this framework, each alternative is represented as a vector of normalized KPI dimensions, and multidimensional performance is aggregated into a composite representation within a unified metric space. The decision problem is formulated as the maximization of expected composite performance across discrete alternatives under scenario uncertainty.

Formally, this formulation defines a decision rule based on expected composite performance, enabling consistent comparison of alternatives within a multidimensional KPI space under stochastic operating conditions.

To operationalize this theoretical construct, a simulation-based mechanism is developed. Stochastic realizations of KPI values are generated for alternative configurations, enabling expected composite performance to be estimated under environmental variability. This computational translation preserves interpretability while allowing decision logic to remain consistent across scenarios.

The theoretical contribution of the study lies in formalizing the integration of performance management and decision modeling within a unified analytical structure. By conceptualizing omnichannel logistics networks as performance-embedded decision systems and advancing an expected composite performance maximization framework, the study contributes to logistics decision theory and provides a reproducible multi-criteria architecture for decision-making under uncertainty.

This study does not propose a new routing algorithm. Instead, it formalizes a KPI-based decision architecture in which performance indicators participate directly in the comparison of alternatives under uncertainty.

## 2 LITERATURE REVIEW AND THEORETICAL BACKGROUND

### 2.1 OPTIMIZATION-ORIENTED LOGISTICS DECISION MODELS

Research on omnichannel logistics decision-making has developed primarily along an optimization-oriented trajectory, focusing on formal network design, facility allocation, and routing configuration under predefined objective structures (Li and Shi, 2024). Existing studies have explored logistics decision-making from different perspectives. One stream of research takes optimization models as the primary approach, improving the operational efficiency of logistics systems through mathematical programming and operations research methods. Such studies typically focus on logistics network design and route configuration, with cost, time, and resource utilization as the main opti-

mization objectives. Nguyen et al., starting from the design of omnichannel distribution networks, combined delivery responsiveness with profit maximization, and constructed a stochastic programming model to characterize network structure decisions (Nguyen et al., 2025); Lin et al., from the perspective of facility location and fulfillment coordination, jointly optimized logistics system layout and fulfillment strategies by considering transportation costs, market demand, and customer channel choices (Lin et al., 2022); Qiu et al. focused on inventory replenishment and transportation coordination, studying distribution routing and replenishment decisions under demand uncertainty and service level constraints (Qiu et al., 2025); Li and Shi adopted a multi-objective perspective, incorporating delivery time, logistics cost, and product characteristics (such as freshness of perishable goods) into a unified optimization framework (Li and Shi, 2024).

## 2.2 KPI and Performance Management Approaches

Another stream of research focuses on performance management tools and emphasizes the role of performance indicators in logistics management. These studies typically use performance measurement frameworks or indicator systems to systematically evaluate logistics operational outcomes (Govindan et al., 2022). Relevant research shows that performance indicators help describe the operational status of logistics systems and support the decomposition and monitoring of management objectives (Özkanlısoy and Bulutlar, 2023). With the development of research, multi-criteria decision-making methods have gradually been introduced into the logistics field to consider multiple performance dimensions simultaneously (Ismail et al., 2024; Więckowski and Sałabun, 2025). Such methods attempt to establish trade-offs among different indicators to support the comparison and ranking of alternatives.

## 2.3 Methodological Discontinuity Between Performance and Decision Logic

These channels serve different demands and correspond to different performance objectives (Cao et al., 2025b). Decision-makers within logistics networks must make trade-offs among cost, service level, and operational stability (Cao et al., 2025a;b). The coexistence of multiple objectives and the coordination of multiple actors make logistics and supply chain systems exhibit characteristics of complex networks. The coupling relationships among different factors significantly increase the complexity of the decision-making process, making traditional single-indicator-oriented decision methods difficult to apply effectively (Ma et al., 2024).

In actual operations, enterprise decision-making typically relies on existing performance management systems and indicator rules. These rules are formed based on historical experience and established managerial logic and tend to remain stable over a long period, but they may be difficult to adjust and optimize in a timely manner under dynamic environmental changes (Munmun et al., 2023). At the same time, key performance indicators are usually used for ex-post performance evaluation; decision-makers rely more on them to summarize and provide feedback on operational outcomes rather than embedding them directly into the process of selecting alternatives, which leads to a certain degree of separation between performance information and the decision-making process (Filani et al., 2023; Rahman et al., 2025). If performance indicators are not incorporated into the decision expression, different alternatives lack a basis for comparability and are difficult to evaluate and select systematically within a multidimensional performance framework, which further weakens the rationality of decision-making (Petropoulos et al., 2026).

Some studies simulate the operational state of supply chains under demand fluctuations, unexpected disruptions, and external risk shocks through scenario design and uncertainty analysis, thereby evaluating the resilience and robustness of different decision alternatives in complex environments (Dolgui et al., 2018; Ivanov, 2022). However, from the overall development of the literature, most existing research still primarily uses performance indicators for ex-post evaluation. Indicator results are typically employed to assess the operational effectiveness of predetermined alternatives, rather than being directly incorporated into the decision modeling process. This results in a certain degree of separation between performance management and decision optimization and, to some extent, limits the systematic application of multi-indicator information at the decision stage. The persistence of this separation suggests a structural misalignment between evaluation architectures and decision architectures, particularly in multidimensional logistics systems.

This unresolved discontinuity motivates the development of a KPI-embedded decision framework.

## 3 CONCEPTUAL FRAMEWORK

Based on the above background, this study proposes a KPI-driven decision optimization framework for omnichannel logistics networks, in which performance indicators are incorporated into the initial decision structure and integrated under a unified weighting scheme to form a multi-indicator decision expression, allowing performance across different dimensions to be compared and selected within the same structural framework.

On this basis, a simulation-based implementation pathway is constructed to translate the decision structure into a computational process. By generating indicator distributions under route-specific scenarios and performing normalization and weighting integration, a repeatable multi-indicator decision ranking mechanism is established.

## 4 METHOD

In the KPI-driven decision framework, KPIs are incorporated as foundational dimensions of the decision model and enter the analytical structure at the initial stage. This enables alternative decision objects to be described and compared within a unified representation. Considering the coexistence of multiple actors and operating conditions in omnichannel logistics networks, the study adopts a KPI-centered methodical structure in which performance dimensions are organized at the modeling stage to ensure completeness and comparability of inputs. Indicators are structured into three reference dimensions: economic (resource consumption), operational (efficiency), and risk-related (system stability), allowing multidimensional performance to be consistently embedded into a common decision space. This dimensional structure reflects the multidimensional nature of logistics performance and enables alternatives to be represented consistently within the decision space.

In practical decision environments, the composition of KPIs varies across scenarios. The three reference dimensions provide a structural basis that allows scenario specific indicators to be incorporated into a unified evaluation framework while maintaining adaptability under changing operating conditions. To operationalize the proposed framework under uncertainty, a simulation-based implementation is developed in which representative transport routes are defined as decision objects. Probabilistic distributions of key performance indicators, including cost, delivery time, emissions, customs efficiency, and transport distance, are specified to capture variability across operating scenarios. These distributions represent typical operational fluctuations observed in logistics environments and allow uncertainty to be represented in a controlled and reproducible manner across simulation scenarios. This enables the decision structure to incorporate environmental uncertainty directly rather than relying on deterministic parameters.

All indicators are then normalized to ensure comparability across different measurement scales:

$$U_i = \sum_{j=1}^{m} w_j \hat{x}_{ij}$$

where $\hat{x}_{ij}$ denotes the normalized value of indicator $j$ for alternative $i$, and $w_j$ represents the corresponding weight.

Normalization transforms heterogeneous indicators into a common scale so that different KPI dimensions can be consistently represented and compared within a unified evaluation space. Cost-type indicators are inversely transformed ("smaller is better"), whereas benefit-type indicators are positively normalized ("larger is better"), allowing all KPIs to enter a unified evaluation domain. A predefined weighting structure is then applied to the normalized indicators to compute a composite performance score. The weighted aggregation represents a linear composite performance approximation within a normalized KPI space. This formulation provides a transparent and interpretable representation of multidimensional performance while preserving the relative contribution of each KPI dimension within the decision model.

The weighting structure reflects the relative importance of performance dimensions in the decision context and allows preference priorities to be expressed across indicators. In the present study, weights are assigned to reflect typical decision priorities in logistics planning environments, where

economic efficiency, operational performance, and system stability jointly influence route selection. This additive formulation assumes partial compensability among KPI dimensions, meaning that strong performance on one indicator may offset weaker performance on another. Such an assumption is consistent with utility-based multi-criteria decision models when trade-offs among performance dimensions are considered acceptable within the decision environment.

By repeating the simulation across multiple stochastic scenarios, the mean composite score of each alternative is calculated to obtain a robust ranking outcome. The use of repeated stochastic realizations allows the framework to evaluate the stability of alternative rankings under varying operational conditions.

Through this process, the conceptual KPI-driven decision structure is translated into a computational decision mechanism capable of continuous operation under uncertain conditions, while maintaining a consistent comparison logic across scenarios.

## 5 RESULTS

The KPI-driven decision framework was applied to evaluate alternative transport routes under stochastic operating conditions. Three route configurations were considered as decision alternatives: the Kazakhstan route, the Kyrgyzstan route, and a hybrid route combining elements of both corridors. To represent environmental variability, Monte Carlo simulation with 1,000 stochastic realizations was conducted for each route configuration. KPI values for transportation cost, delivery time, emissions, customs efficiency, and transport distance were generated using stochastic distributions reflecting typical operational variability in cross-border logistics environments.

To ensure cross-indicator comparability, all KPI dimensions were normalized onto a unified [0,1] scale. Cost-type indicators were inversely transformed, while benefit-type indicators were positively normalized. The normalized indicators entered a weighted aggregation structure, yielding the composite performance index:

$$U_i = \sum_{j=1}^{m} w_j \hat{x}_{ij}$$

Under scenario uncertainty, the decision rule is defined as the maximization of expected composite performance:

$$R^* = \underset{i}{\mathrm{argmax}}\, \mathbb{E}\left[U_i\right]$$

The expected value is estimated through Monte Carlo simulation across $N$ stochastic realizations:

$$\overline{CCI}_i = \frac{1}{N} \sum_{s=1}^{N} U_i^{(s)}$$

where $U_i^{(s)}$ denotes the composite performance of alternative $i$ in scenario $s$.

The simulation results show that although individual KPI values fluctuate across stochastic realizations, the aggregated composite index maintains a consistent comparative structure across route alternatives. This indicates that the KPI-driven decision formulation captures structural differences in multidimensional performance rather than scenario-specific variations.

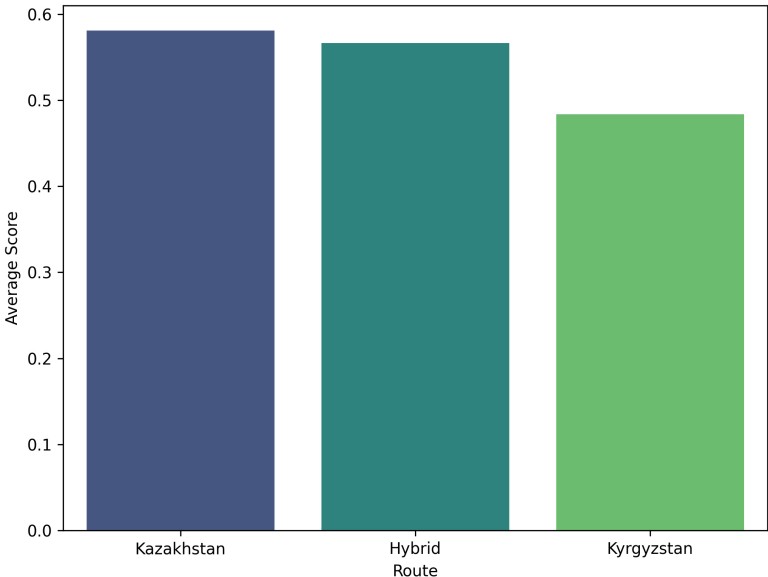

Figure 1: Average composite performance of route alternatives.

Figure 1 presents the mean composite scores obtained for each route across the simulation runs. The results indicate systematic differences in aggregated KPI outcomes among the alternatives.

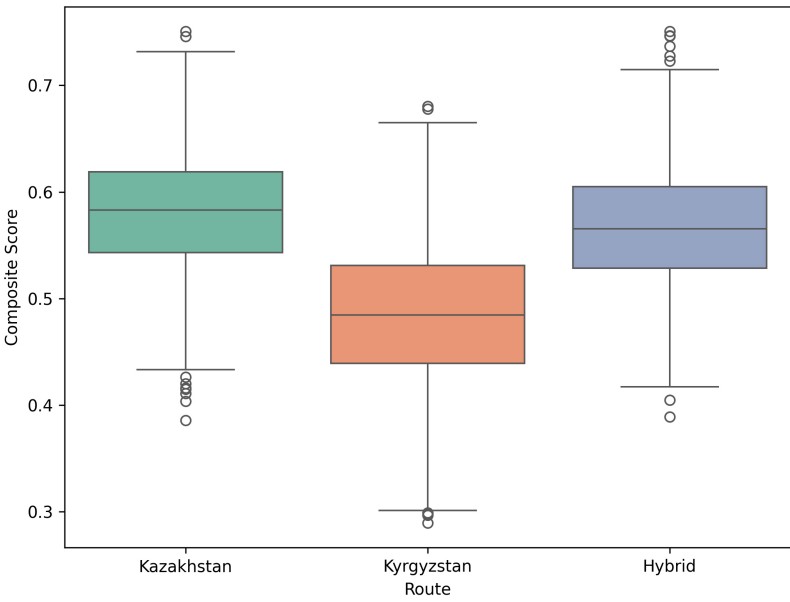

Figure 2: Distribution of composite performance by route

Figure 2 illustrates the distribution of composite scores across the simulation runs. Although KPI values vary across stochastic realizations, the distribution ranges of the three routes remain sufficiently separated to preserve a stable ranking structure.

These results provide quantitative evidence that the KPI-driven decision formulation supports consistent alternative comparison under uncertain operating conditions.

## 6  DISCUSSION

Conceptually, the proposed framework reframes logistics decision-making as a performance-structured decision architecture. Rather than treating key performance indicators (KPIs) as external evaluation outputs applied after a solution is selected, the framework incorporates KPI dimensions directly into the decision formulation. In this representation, alternatives are evaluated within a multidimensional KPI space, allowing performance information to participate directly in the decision rule under uncertainty. Our prior study on AI-enabled route selection in a Russian–Chinese bioethanol corridor (Barykin et al., 2025) demonstrated the feasibility of multi-criteria logistics optimization using AI-based methods. The present study extends this line of work by formalizing KPI dimensions as structural components of the decision model rather than as purely evaluative indicators.

This formulation links performance measurement with decision modeling. In conventional logistics practice, KPIs are mainly used to summarize operational outcomes and support managerial monitoring. Decision models, in contrast, define how alternatives are compared and selected. By embedding normalized KPIs and their relative priorities within the decision rule, the framework allows multidimensional performance information to directly influence alternative comparison within a unified decision structure.

The simulation-based implementation further illustrates the behavior of the framework under uncertainty. Although individual KPI values fluctuate across stochastic scenarios, the normalization and aggregation structure remains constant, preserving comparability across realizations. The simulation results indicate that the relative ranking of alternatives remains stable across repeated scenarios, suggesting that the decision outcomes are robust to moderate variations in KPI realizations.

Several limitations should be acknowledged. First, the weighted additive formulation assumes partial compensability among KPI dimensions and is most appropriate when trade-offs between indicators are acceptable in the decision context. Second, the framework is designed primarily for discrete alternative comparison rather than continuous network optimization with endogenous decision variables. In many logistics planning contexts, mathematical programming methods can be used to generate feasible network configurations or candidate routing solutions. The KPI-driven framework proposed in this study is intended to complement such optimization approaches by providing a transparent mechanism for comparing alternatives when multiple performance dimensions and uncertainty must be considered simultaneously. Third, decision outcomes depend on KPI specification, normalization ranges, and weight assignment; therefore, sensitivity checks remain advisable when applying the framework to new environments.

Future research may extend the framework in several directions. Empirical validation using real operational data would strengthen external validity and allow scenario distributions to be estimated from observed variability. Alternative weight identification methods and scenario-dependent weighting schemes could also be explored to reflect evolving managerial priorities. In addition, integration with real-time data environments may enable the framework to support adaptive decision processes while preserving interpretability.

Compared with simpler decision approaches based on single-indicator optimization or deterministic evaluation, the proposed framework provides a structured mechanism for integrating multiple KPI dimensions and uncertainty within a unified decision architecture. In contrast to MCDM sustainability frameworks that focus primarily on performance evaluation (e.g., (Ismail et al., 2024)), the present study embeds KPI structures directly within the decision rule, enabling multidimensional performance to shape alternative selection under uncertainty.

## 7  CONCLUSION

This study develops a KPI-driven framework for decision optimization in omnichannel logistics networks by embedding multidimensional performance directly into the **decision formation process** rather than limiting KPIs to ex-post evaluation. By treating KPIs as endogenous decision dimensions, the proposed approach converts performance information into a structured comparison logic that supports systematic alternative selection under uncertain and changing operating conditions.

A simulation-based operationalization translates the conceptual structure into a computational decision mechanism. Monte Carlo scenario generation allows environmental variability to be reflected in KPI realizations, while invariant normalization and aggregation rules preserve comparability across scenarios. The empirical demonstration shows that the resulting composite performance formulation produces stable ranking outcomes based on expected multidimensional performance across stochastic scenarios.

The framework contributes to logistics decision theory by bridging performance management and decision modeling within a unified analytical structure and by providing an interpretable multi-criteria architecture suitable for discrete alternative selection under uncertainty. For transparency and reproducibility, the model is operationalized through a Python-based simulation implementation provided as supplementary material, enabling verification and facilitating further methodological development and extension to additional omnichannel settings

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
