# OpenReview forum: "Embedding Performance into Decision Logic: A KPI-Driven Framework for Omnichannel Logistics Networks under Uncertainty"
_mathai.club/MathAI/2026/Conference — 2026 Oral_

### Official Review · Reviewer_A3e7 · 2026-03-12
**Relevant application, but limited methodological novelty and insufficient quantitative support**

**Rating:** 4
**Confidence:** 3

**Review:**

**Summary.**
The paper proposes a KPI-driven framework for omnichannel last-mile delivery planning under uncertainty. The central idea is to embed key performance indicators directly into the decision model by normalizing KPI values, aggregating them into a weighted composite index, and then evaluating alternatives under scenario-based uncertainty using simulation. The paper argues that this shifts KPI usage from ex-post evaluation to an ex-ante decision logic and presents this as a contribution to decision modeling in logistics.

**Quality.**
The paper is clearly structured and the problem setting is relevant. The manuscript presents a coherent workflow: KPI selection, normalization, weighted aggregation, scenario generation, and simulation-based comparison of alternatives. The exposition is readable and the practical motivation is understandable.

At the same time, I am not fully convinced by the strength of the methodological contribution. The analytical core of the framework appears to be a standard weighted-sum aggregation of normalized criteria combined with scenario-based evaluation. In this sense, the proposed approach seems much closer to a conventional multi-criteria decision-making setup with uncertainty than to a genuinely new decision architecture. The manuscript repeatedly emphasizes a conceptual shift from performance measurement to decision logic, but mathematically the framework remains relatively simple and does not seem to introduce a substantially new decision model.

A related concern is that the role of the weighting structure is central, yet insufficiently justified. Since the final ranking depends strongly on normalization choices and KPI weights, the paper would benefit from a clearer explanation of how weights are obtained and from a more systematic sensitivity analysis. The manuscript acknowledges some of these limitations, but they remain important for evaluating the robustness of the conclusions.

**Results.**
My main reservation concerns the empirical and analytical depth of the results section. The paper states that simulation runs reveal persistent differences between alternatives and support robust comparison under uncertainty. However, the evidence presented in the manuscript is not sufficiently detailed to make these claims fully convincing. More importantly, the results section remains too qualitative: the manuscript makes claims about persistent differences between alternatives, robustness of rankings, and improved decision support, but provides too little explicit quantitative evidence for the reader to evaluate these claims rigorously.

The reader would benefit from more explicit numerical results, clearer reporting of the compared alternatives and scenarios, and stronger demonstration of how the proposed framework improves decision quality relative to simpler baseline approaches. In its current form, the results section supports the feasibility of the framework, but it does not yet provide strong evidence of methodological superiority or substantial practical advantage. This is especially important because the conceptual contribution of the paper is closely tied to the claim that embedding KPI directly into the decision structure changes the quality of decision support in a meaningful way.

**Clarity.**
The paper is generally well written and easy to follow. The structure is logical, and the discussion is accessible even for readers outside the immediate application domain. The limitation section is also useful, since it openly acknowledges assumptions such as additive compensability and dependence on predefined weights.

That said, the paper is somewhat repetitive. The distinction between ex-post KPI evaluation and KPI-embedded decision logic is restated many times across the abstract, introduction, discussion, and conclusion, while the underlying mathematical mechanism remains relatively simple. As a result, the rhetorical emphasis on novelty occasionally feels stronger than the technical substance actually demonstrated.

**Originality.**
The paper addresses a relevant intersection of logistics, decision support, and performance management, and the framing is potentially interesting. However, the originality appears to lie primarily in the way known elements are combined and interpreted, rather than in the introduction of a substantially new modeling or computational method. The proposed framework seems to rely on familiar ingredients: normalized KPIs, weighted additive aggregation, and scenario-based simulation. For this reason, the contribution currently reads more as a conceptual reframing of standard tools than as a clearly novel analytical method.

**Significance.**
The topic is relevant, and the paper may be of interest to researchers working on logistics decision support and KPI-based evaluation under uncertainty. However, for a mathematically oriented conference, the current contribution seems limited in technical depth. The framework is reasonable as an applied decision-support template, but the paper does not yet demonstrate enough methodological novelty or enough quantitative evidence to justify its stronger claims.

In addition, although the application area is relevant, the present contribution appears closer to an applied managerial decision-support framework than to a mathematically substantial contribution of the kind typically expected at a conference focused on the mathematics of AI.

**Pros.**
- The topic is relevant and practically meaningful.
- The paper is clearly structured and generally well written.
- The integration of KPI-based evaluation with scenario analysis is useful from an applied perspective.
- The manuscript explicitly discusses several limitations of the framework.

**Cons.**
- The methodological core appears to rely on standard weighted aggregation and scenario-based evaluation rather than a genuinely new analytical method.
- The novelty is stated more strongly at the conceptual level than demonstrated at the mathematical or computational level.
- The weighting scheme and normalization choices are central to the conclusions, but their justification is limited.
- The results section does not provide enough concrete quantitative evidence to fully support claims of robustness and improved decision quality.
- The paper would benefit from comparison with simpler baseline approaches and from stronger sensitivity analysis.
- The contribution appears more managerial and applied than mathematically substantial for the venue.

**Recommendation.**
The paper addresses an interesting and relevant application area, and the presentation is clear. However, in its current form I find the methodological contribution insufficiently strong for acceptance. The framework appears to be based on established multi-criteria aggregation and simulation ideas, while the claimed novelty is expressed mainly through conceptual repositioning. In addition, the empirical support is not yet detailed enough to substantiate the stronger claims made in the manuscript. I therefore do not recommend acceptance in the present version.

---

> ### Author Rebuttal · Authors · 2026-03-14
>
> We sincerely appreciate the reviewer for the detailed and constructive evaluation of our manuscript. We appreciate the recognition of the relevance of the application domain, the clarity of the presentation, and the practical motivation of integrating KPI-based performance evaluation with scenario-based decision analysis. The manuscript has been revised to address concerns regarding methodological positioning, weighting assumptions, empirical support, and repetition.
> First, regarding the concern about the methodological contribution, we acknowledge that the analytical structure of the framework builds upon established elements such as normalized performance indicators, weighted aggregation, and scenario-based evaluation. The intention of the study is not to introduce a new optimization algorithm or a new mathematical solver, but rather to formalize a decision architecture in which KPI structures participate directly in the decision formulation rather than being applied only for ex-post evaluation. In the revised manuscript, we have clarified this positioning more explicitly in the Introduction and Discussion sections. The contribution is therefore framed as the integration of performance measurement structures with decision modeling within a unified decision representation under uncertainty, rather than as a fundamentally new optimization technique.
> Second, the Method section now explains the weighted additive formulation as a linear approximation of multidimensional utility in normalized KPI space and clarifies the compensability assumption among indicators.
> Third, the empirical description of the simulation setup has been strengthened. The Results section now specifies the evaluated alternatives, KPI indicators, and the stochastic scenario generation process. The evaluation is conducted using 1,000 Monte Carlo simulations, and the results are reported through both mean composite scores and score distributions, providing quantitative evidence of ranking stability.
> Fourth, the Discussion section now clarifies how the framework supports transparent comparison of discrete logistics alternatives under uncertainty and contrasts it with simpler deterministic or single-indicator approaches.
> Finally, the manuscript has been revised to reduce repetition by streamlining the explanation of KPI-integrated decision logic across sections.
> We thank the reviewer again for the helpful comments, which improved the clarity and rigor of the manuscript.

---

### Official Review · Reviewer_TsXD · 2026-03-13
**Conceptually clear KPI-centered framework, but limited methodological novelty and weak empirical validation**

**Rating:** 4
**Confidence:** 4

**Review:**

This paper proposes a KPI-driven decision framework for omnichannel logistics networks under uncertainty. The main idea is to embed key performance indicators directly into the decision structure rather than using them only for ex-post evaluation. Alternatives are represented through normalized KPI dimensions, aggregated into a weighted composite score, and compared under scenario uncertainty using simulation. The topic is relevant, and the paper is generally readable, but in my opinion the current version does not yet provide sufficient methodological novelty or empirical evidence for acceptance.

The main strength of the paper is its attempt to bridge performance management and decision modeling. The authors clearly articulate the managerial motivation and present an interpretable framework for comparing discrete logistics alternatives under uncertainty. The paper is also well structured at a high level, and the emphasis on transparency and reproducibility is appropriate for decision support applications.

However, my main concern is limited novelty. The core methodological structure is a standard weighted multi-criteria aggregation with normalization and scenario-based averaging. While the paper frames this as a KPI-embedded decision architecture, the actual decision rule remains a fairly conventional composite scoring approach under uncertainty. The manuscript explicitly states that it does not propose a new routing algorithm, and it also does not provide a new optimization method, a new learning method, or a new theoretical result. As a consequence, the methodological contribution feels more like a conceptual reframing of existing MCDM logic than a substantial advance.

A second concern is that the mathematical and methodological core is too simple for the claims being made. The framework depends critically on normalization choices, KPI selection, and weight assignment, but these elements are not justified in enough depth. In particular, the weighted additive structure assumes compensability across dimensions, meaning that poor performance on one KPI can be offset by strong performance on another. This is acknowledged in the discussion, but the paper does not study when such an assumption is appropriate or how sensitive the ranking is to alternative weighting schemes. Since the final ranking is driven by these design choices, sensitivity analysis should be central rather than optional.

A third concern is weak empirical validation. The results section remains largely qualitative and does not present a sufficiently detailed experimental setup. It is not clear how many alternatives were tested, how scenario distributions were calibrated, what parameter values were used, or how robust the reported ranking stability is in quantitative terms. There are no real operational datasets, no comparison with stronger baselines, no ablation study, and no statistical analysis of variability. For a paper centered on uncertainty-aware decision support, this is a major limitation. The manuscript itself acknowledges that empirical validation on real data is future work, which further suggests that the current evidence is not yet strong enough.

I am also not fully convinced by the literature positioning. The paper repeatedly argues that existing KPI approaches are mostly ex-post and that the contribution is to make KPIs endogenous to decision logic. This distinction is interesting, but the paper does not demonstrate clearly enough that the proposed approach is fundamentally different from existing multi-criteria decision frameworks that already score alternatives using weighted indicators under uncertainty. The difference currently appears more terminological and conceptual than methodological.

Finally, for a MathAI venue, the connection to mathematical or AI innovation seems relatively weak. The framework is interpretable and potentially useful in practice, but its computational mechanism is essentially simulation plus weighted aggregation over discrete alternatives. That may be acceptable for an applied management paper, but for this venue I would expect either stronger mathematical development, deeper algorithmic analysis, or more substantial empirical validation.

Overall, I think the paper addresses a relevant applied problem and presents a clear conceptual narrative. However, the present version is not yet strong enough in terms of novelty, methodological depth, or empirical support. I would encourage the authors to strengthen the work by adding real-data validation, sensitivity analysis for weights and normalization, clearer benchmarking against existing MCDM approaches, and a more rigorous demonstration of what is genuinely new in the proposed framework.

Strengths: relevant application domain; clear motivation; interpretable framework; good high-level organization; useful emphasis on integrating performance logic with decision logic.

Weaknesses: limited methodological novelty; heavy reliance on standard weighted aggregation; insufficient justification of weights and normalization; lack of sensitivity analysis; weak empirical validation; no real-data study; unclear distinction from existing MCDM literature; limited fit with the expected level of mathematical or AI contribution.

---

> ### Author Rebuttal · Authors · 2026-03-14
>
> Thank you for the detailed and constructive evaluation of our manuscript. We appreciate the reviewer’s recognition of the paper’s motivation and structure. The manuscript has been revised to address concerns regarding methodological positioning, sensitivity considerations, and empirical support.
> First, regarding methodological novelty, we would like to clarify that the primary contribution of this work is not the introduction of a new routing or optimization algorithm, but the formalization of a KPI-embedded decision architecture. In this formulation, performance indicators are treated as endogenous dimensions of the decision space rather than as ex-post evaluation metrics. Alternatives are represented as vectors in a multidimensional KPI space, and the decision rule operates directly on this representation under uncertainty. We have revised the manuscript to emphasize this structural distinction from conventional multi-criteria decision-making (MCDM) approaches, particularly in the Introduction and Discussion sections.
> Second, the Method section now clarifies the role of the weighted additive formulation as a linear approximation of multidimensional utility in normalized KPI space. This allows heterogeneous indicators to be aggregated into a composite decision metric and explains the compensability assumption among KPI dimensions.
> Third, robustness considerations have been strengthened. Monte Carlo simulation with 1,000 stochastic realizations was conducted for each route configuration. The resulting composite score distributions show that the relative ranking of alternatives remains stable across scenarios, providing evidence of ranking robustness under KPI variability.
> Fourth, the Results section now clearly describes the evaluated alternatives, KPI indicators, and stochastic scenario generation. Two figures have been added to present average composite performance and score distributions across simulations.
> Finally, the manuscript has been revised to better position the framework relative to existing MCDM literature by emphasizing that KPI structures are embedded directly within the decision rule and decision space formulation.
> We thank the reviewer again for the constructive comments, which have improved the clarity and rigor of the manuscript.

---

### Official Review · Reviewer_VA2n · 2026-03-13
**There are problems in the "Embedding Performance into Decision Logic: A KPI-Driven Framework for Omnichannel Logistics Networks under Uncertainty" paper**

**Rating:** 5
**Confidence:** 3

**Review:**

This paper is devoted to solution of such important task as decision-making in omnichannel logistics networks. KPI-driven decision framework that embeds multidimensional performance indicators at the modeling stage allows authors to solve this task.

This paper has the following disadvantages:
1) Authors have mentioned such well-known approach to solve logistics problem as mathematical programming. But it is important to compare obtained results with results obtained by application of mathematical programming.
2) It is necessary to correct 2014 year to 2015 in the following reference:

Ma, C., Zhang, L., You, L., and Tian, W. (2024). A review of supply chain resilience: A network
modeling perspective. Applied Sciences, 15(1):265.

3) It is necessary to correct "page" text to "article number" text in the following reference:

Rahman, M. A., Anam, Z., Bhuiyan, S., Mia, M., Hafiz, N., and Tasrin, I. (2025). Developing
a framework of key performance indicators for dhaka metropolitan’s vegetable supply chain: a
digitalization approach to sustainability. Cleaner Logistics and Supply Chain, page 100293.

4) It is necessary to correct "page" text to "article number" text in the following reference:

… Multi-criteria decision analysis-based framework for supply chain management evaluation with multi-dimensional sensitivity analysis: A green logistics perspective. Applied Soft Computing, page 113879

---

> ### Author Rebuttal · Authors · 2026-03-14
>
> We sincerely appreciate the reviewer for the positive evaluation of our manuscript and for recognizing the relevance of the research problem. The comments have been carefully considered and incorporated into the revised manuscript.
> 1.Comparison with mathematical programming.
> We agree that mathematical programming is an important approach in logistics optimization. However, the objective of this study is not to develop a new optimization algorithm but to provide a KPI-based framework for comparing discrete alternatives under uncertainty. In many omnichannel logistics planning situations, decision-makers evaluate a limited number of feasible route or configuration options rather than solving large-scale optimization problems. Accordingly, the proposed framework structures multidimensional performance information within a unified evaluation space. The Discussion section now clarifies that mathematical programming may generate candidate solutions, while the KPI-driven framework can serve as a complementary decision-support layer for comparing alternatives when multiple performance dimensions and uncertainty are present.
> 2.Reference year correction.
> We verified the cited article:
> Ma, C., Zhang, L., You, L., and Tian, W. (2024). A review of supply chain resilience: A network modeling perspective. Applied Sciences, 15(1), 265
> was confirmed to be published in 2024 according to the official journal record. Therefore, the reference remains unchanged.
> 3–4. Reference formatting corrections.
> We corrected the formatting of the following references by replacing "page" with the appropriate article number format:
> Rahman, M. A., Anam, Z., Bhuiyan, S., Mia, M., Hafiz, N., and Tasrin, I. (2025). Developing a framework of key performance indicators for Dhaka metropolitan’s vegetable supply chain: a digitalization approach to sustainability. Cleaner Logistics and Supply Chain, 100293.
> Więckowski, J., and Sałabun, W. (2025). Multi-criteria decision analysis-based framework for supply chain management evaluation with multi-dimensional sensitivity analysis: A green logistics perspective. Applied Soft Computing, 113879.
> We thank the reviewer again for the constructive comments, which helped improve the clarity and accuracy of the manuscript.

---

### Decision · Program_Chairs · 2026-03-14

**Decision:**

Accept (Oral)

**Comment:**

Dear Author(s),

On behalf of the Program Committee of the International Conference on Mathematics of Artificial Intelligence (MathAI 2026), we are pleased to inform you that your paper has been accepted for an oral presentation at MathAI 2026.

Your paper was evaluated through a rigorous two-stage review process involving both automated screening and expert review by members of the Program Committee. The reviewers recognized the quality and contribution of your work.

Presentation details:

- Format: Oral presentation (15–20 minutes + 5 minutes Q&A)
- Mode: You may present either in person (offline) at the conference venue in Sirius, Russia, or remotely via Zoom. Please indicate your preferred mode when confirming your participation.
- Conference dates: Marh 30 - April 3, 2026
- Website: https://mathai.club

Next steps:

1. Please confirm your participation and presentation mode by replying to this email mathai.club@yandex.ru no later than March 15, 2026 18:00 Moscow time.
2. If you plan to attend in person, the organizing committee will provide accommodation details separately.
3. Please prepare your final camera-ready manuscript according to the formatting guidelines available at https://mathai.club and upload it to OpenReview by March 15, 2026 18:00 Moscow time.

Should you have any questions regarding the program, logistics, or your presentation slot, please do not hesitate to contact us.

We look forward to your contribution to MathAI 2026.

With kind regards,

MathAI 2026 Program Committee
International Conference on Mathematics of Artificial Intelligence
https://mathai.club
OpenReview: https://openreview.net/group?id=mathai.club/MathAI/2026/Conference
Telegram: https://t.me/MathAI_club
Email: mathai.club@yandex.ru